# Association between Homocysteine and Vitamin D Levels in Asymptomatic Korean Adults

**DOI:** 10.3390/nu16081155

**Published:** 2024-04-13

**Authors:** Yun-Ah Lee, Sung-Goo Kang, Sang-Wook Song, Se-Hong Kim

**Affiliations:** Department of Family Medicine, St. Vincent’s Hospital, College of Medicine, The Catholic University of Korea, Seoul 16247, Republic of Korea

**Keywords:** homocysteine, vitamin D, cardiovascular disease

## Abstract

An increased homocysteine level is a risk factor for cardiovascular disease, venous thromboembolism, cerebrovascular disease, and chronic kidney disease. In addition, vitamin D deficiency is associated with coronary artery disease and metabolic disorders. The present study included data from 1375 adults (895 men and 480 women) with a mean age of 52.62 ± 9.94 years who visited the Health Promotion Center of the University Hospital in Gyeonggi-do, Republic of Korea from January 2018 to December 2022 for routine checkups that included assessments of their homocysteine and vitamin D levels. Homocysteine levels were positively associated with age, a history of hypertension, a history of diabetes, current smoking habits, and levels of low-density lipoprotein cholesterol, creatinine, uric acid, and high-sensitivity C-reactive protein. By contrast, vitamin D levels were negatively associated with serum levels of homocysteine after adjusting for covariates (*β* = −0.033, *p* < 0.001). Additional long-term prospective studies are needed to elucidate the presence of a causal relationship between vitamin D status and serum levels of homocysteine in asymptomatic Korean adults. An intervention trial is warranted to determine whether the administration of vitamin D is helpful for the primary prevention of cardiovascular disease by lowering the homocysteine level in this population.

## 1. Introduction

Homocysteine is an intermediate metabolite in the methionine metabolism pathway, and several enzymes and vitamins, such as folate, vitamin B6, and vitamin B12, are involved in this complex pathway. The excessive consumption of products providing methionine, deficiencies of vitamins and enzymes involved in homocysteine metabolism, and mutations in genes encoding enzymes involved in methionine metabolism can increase levels of homocysteine, adversely affecting human health [1,2,3,4]. Increased homocysteine levels are associated with pro-thrombotic and pro-oxidant states, leading to endothelial dysfunction and platelet hyper-reactivity [4,5]. Hyperhomocysteinemia is a risk factor for cardiovascular disease, venous thromboembolism, cerebrovascular disease, chronic kidney disease, and congenital disorders [3,6,7,8,9,10]. Furthermore, previous studies revealed that hyperhomocysteinemia is a risk factor for cardiovascular and all-cause mortality [11,12]. A cohort study of 2968 patients with cardiovascular disease revealed that patients with homocysteine concentrations > 15.6 μmol/L showed a nearly three times higher hazard ratio for death than those with homocysteine concentrations < 9.8 μmol/L [12]. Vitamin D is synthesized in sunlight-exposed skin and can be obtained through foods or supplements. Circulating 25-hydroxyvitamin D [25(OH)D] is a nutritional biomarker for vitamin D status [13]. Vitamin D plays an important role in maintaining calcium homeostasis and bone health, and it has been studied for its health benefits, including the prevention of cardiovascular disease [13,14,15]. Previous studies have shown that vitamin D deficiency is associated with the development of coronary artery disease and metabolic disorders, including abnormal blood glucose and lipid profiles and increased homocysteine levels [16,17]. Vitamin D levels were inversely correlated with total cholesterol, low-density lipoprotein cholesterol (LDL-cholesterol), and homocysteine in patients with hyperlipidemia [16], whereas vitamin D was positively associated with high-density lipoprotein cholesterol (HDL-cholesterol). Several studies have shown that both high homocysteine levels and low 25(OH)D concentrations are potential risk factors for cardiovascular disease [18,19]. A study of the National Health and Nutrition Examination Survey data, which included a variety of races, showed a significant inverse relationship between 25(OH)D and homocysteine levels among people with 25(OH)D levels < 21 ng/mL [20]. Several clinical studies have noted a possible interaction between homocysteine and vitamin D levels. Moreover, there has been an interest in these two markers because of their common significance in some pathogenesis related to the prevention and treatment of cardiovascular diseases, of which incidences are growing, rendering them one of the leading causes of death [18,19,20]. However, few studies have evaluated the association between homocysteine and 25(OH)D levels in specific demographic groups, such as the healthy asymptomatic Korean population, which may have distinct genetic, dietary, and environmental factors influencing these nutrient levels and metabolic pathways. Therefore, we studied the relationship between these levels and risk factors for cardiovascular disease in healthy Korean adults.

## 2. Materials and Methods 

### 2.1. Study Population

This cross-sectional study involved 1375 adults aged ≥ 19 years who visited the Health Promotion Center of the University Hospital in Gyeonggi-do, Republic of Korea from January 2018 to December 2022. The participants underwent routine health checkups, which included comprehensive evaluations of their homocysteine and 25(OH)D levels among other health markers. We specifically excluded individuals with a history of cardiovascular or cerebrovascular disease presenting moderate to severe symptoms, patients diagnosed with any cancer, and those under treatment for thyroid dysfunctions, such as hyperthyroidism or hypothyroidism with moderate to severe symptoms. This was a retrospective study of medical records obtained from health examinations conducted within the specified period. The study protocol was approved by the institutional review board of the Catholic University of Korea St. Vincent’s Hospital (IRB approval number: VC 24RISI0032), with a waiver for the requirement for informed consent. All procedures of this study strictly adhered to all applicable guidelines and regulations to ensure ethical compliance and participants’ safety. 

### 2.2. Laboratory Measurements

Participants were required to undergo an 8 h overnight fast before the collection of venous blood samples. An autoanalyzer (Hitachi 747 Autoanalyzer; Hitachi, Tokyo, Japan) was used for the quantitative analysis of serum homocysteine, fasting plasma glucose, LDL-cholesterol, triglycerides, HDL-cholesterol, and LDL-cholesterol levels. Additionally, an autoanalyzer (AU 5822 Automatic Clinical Chemistry Analyzer; Beckman Coulter, Brea, CA, USA) was utilized for creatinine, uric acid, and high-sensitivity C-reactive protein (hs-CRP). Vitamin D (measured as 25-hydroxyvitamin D [25(OH)D]) was measured using Architect i2000 SR (Abbott, North Chicago, IL, USA), and thyroid-stimulating hormone (TSH) was analyzed using an immunoassay analyzer (DXI 800; Beckman Coulter, USA). In addition, an autoanalyzer (D-100; BioRad, Hercules, CA, USA) was used for the analysis of glycated hemoglobin (HbA1c). These measurements provided a comprehensive overview of each participant’s metabolic status and potential risk factors for cardiovascular diseases.

### 2.3. Other Variables

Anthropometric assessments and vital signs were meticulously measured by trained medical personnel following standardized protocols. Blood pressure readings were taken in a seated position after a 10 min rest period, thereby ensuring accuracy and consistency. Waist circumference was measured at the level of the iliac crest, with participants standing and the measurement tape placed in a horizontal plane around the abdomen after exhalation. Height and weight were measured under controlled conditions after an overnight fast, with participants wearing light gowns and no footwear, facilitating the calculation of body mass index (BMI) as the weight in kilograms divided by the square of the height in meters. Furthermore, all study participants were asked to complete a detailed questionnaire that covered their drinking and smoking habits, physical activity levels, medical history, and current medications. The classification of drinking habits considered the frequency per week (i.e., ≤1 time per week or >1 time per week [moderate drinking]). Patient smoking status was categorized as them being a nonsmoker, current smoker, or ex-smoker, and physical activity was quantified based on the frequency of exercise per week (i.e., never exercise, exercise ≤ 3 times per week, or exercise ≥ 4 times per week [physically active]). 

### 2.4. Statistical Analyses

The demographic and clinical characteristics of the study population were analyzed using the independent *t*-test for continuous variables and the chi-square test or Fisher’s exact test for categorical variables, with results presented as a mean ± standard deviation or number (percentage). Pearson’s correlation analysis was used to explore the relationships between serum homocysteine levels and clinical variables, providing insight into potential risk factors and associations. Multiple linear regression analysis was utilized to assess the association between serum homocysteine and 25(OH)D levels, adjusting for a wide array of covariates across three models to ascertain the impact of various factors on these associations. Model 1 was adjusted for basic demographic factors (age and sex); Model 2 for lifestyle and basic health parameters (BMI, waist circumference, systolic blood pressure [SBP], diastolic blood pressure [DBP], smoking, alcohol consumption, and physical activity); and Model 3 for comprehensive health status indicators, including hypertension, diabetes mellitus, dyslipidemia, and specific laboratory measures (fasting glucose, HbA1c, triglycerides, cholesterol levels, creatinine, uric acid, hs-CRP, and TSH). All statistical analyses were performed using SPSS software version 21.0 (IBM Corp., Armonk, NY, USA). *p* values < 0.05 were considered statistically significant.

## 3. Results

### 3.1. Characteristics of the Study Participants 

The study included 1375 participants, comprising 895 men and 480 women, presenting an average age of 52.62 ± 9.94 years (men, 53.23 ± 9.59 years; women, 51.48 ± 10.48 years). A comprehensive evaluation of the demographics and health metrics revealed significant differences between genders in several areas: age, BMI, waist circumference, SBP, DBP, and lifestyle factors such as smoking, alcohol consumption, and exercise habits. Additionally, metabolic and biochemical assessments highlighted differences in fasting glucose, HbA1c, triglycerides, HDL-cholesterol, creatinine, uric acid, hs-CRP, TSH, 25(OH)D, and homocysteine levels (Table 1). Compared to women, men were older and had a higher BMI, SBP, DBP, fasting glucose, triglycerides, uric acid, hs-CRP, and homocysteine, and lower 25(OH)D and HDL-cholesterol.

### 3.2. Correlations between Homocysteine Levels and Clinical Variables

We examined the relationships between homocysteine levels and several clinical variables (Table 2). Homocysteine levels were positively correlated with age, BMI, waist circumference, SBP, DBP, fasting glucose, HbA1c, triglycerides, LDL-cholesterol, creatinine, uric acid, and hs-CRP. Notably, an inverse correlation was identified between homocysteine levels and both 25(OH)D (*r* = −0.098, *p* < 0.001) and HDL-cholesterol levels (*r* = −0.142, *p* < 0.001), indicating that higher 25(OH)D and HDL-cholesterol levels are associated with lower homocysteine levels.

### 3.3. Associations between Serum Levels of Homocysteine and Vitamin D 

Figure 1 shows an inverse linear relationship between 25(OH)D and homocysteine levels (*r*= −0.098, *p* < 0.001). The cross-sectional associations between serum homocysteine and 25(OH)D concentrations are presented in Table 3. When all 1375 participants were considered, a multivariable linear regression analysis revealed an inverse association between homocysteine and 25(OH)D levels after adjusting for age, sex, BMI, waist circumference, SBP, DBP, smoking, alcohol, exercise, hypertension, diabetes mellitus, dyslipidemia, fasting glucose, HbA1c, triglycerides, HDL-cholesterol, LDL-cholesterol, creatinine, uric acid, hs-CRP, and TSH (*β* = −0.033, *p* < 0.001) (Table 3).

### 3.4. Multiple Linear Regression Analysis of Homocysteine Levels and Clinical Variables 

A multiple regression analysis exploring the influence of clinical variables on homocysteine levels identified several significant associations. Age, history of hypertension, and history of diabetes were positively associated with homocysteine levels (*β* = 0.025, *p* = 0.016; *β* = 0.876, *p* < 0.001; and *β* = 1.480, *p* < 0.001, respectively). Lifestyle factors like current smoking status, as well as other clinical variables including LDL-cholesterol, creatinine, uric acid, and hs-CRP were also positively associated with homocysteine concentrations (*β* = 0.892, *p* = 0.001; *β* = 0.009, *p* = 0.002; *β* = 6.788, *p* < 0.001; *β* = 0.245, *p* = 0.002; and *β* = 0.687, *p* = 0.016, respectively), underscoring the complex interplay between lifestyle, metabolic health, and homocysteine concentrations. By contrast, an increase in the 25(OH)D level was associated with a significant decrease in homocysteine level (*β* = −0.033, *p* < 0.001), suggesting that adequate vitamin D status could play a protective role against elevated homocysteine levels, potentially mitigating related health risks (Table 4). However, we found no significant differences in BMI, waist circumference, SBP, DBP, dyslipidemia, alcohol consumption, exercise, fasting glucose, HbA1c, triglycerides, HDL-cholesterol, and TSH levels. This multifaceted analysis provides an advanced understanding of the factors influencing homocysteine levels in a healthy Korean adult population. The significant findings regarding the associations between homocysteine levels, vitamin D status, and various lifestyle and metabolic factors underscore the importance of a holistic approach to managing cardiovascular risk factors.

## 4. Discussion

The associations between serum levels of homocysteine and 25(OH)D and several risk factors for cardiovascular disease were investigated in asymptomatic Korean adults. Age, a history of hypertension, a history of diabetes, current smoking status, and levels of LDL-cholesterol, creatinine, uric acid, and hs-CRP were positively associated with homocysteine levels, whereas 25(OH)D levels were negatively associated with homocysteine levels after adjusting for covariates.

Previous studies have revealed mixed findings regarding the associations between homocysteine and 25(OH)D levels and many clinical variables [18,20,21,22]. A study of 3150 patients who underwent coronary angiography showed an independent inverse relationship between 25(OH)D and homocysteine levels [18]. In addition, the association between homocysteine levels and the prevalence and severity of coronary artery disease was significant only in patients with lower 25(OH)D levels (<14.6 mg/dL). In a randomized, double-blind, placebo-controlled clinical trial of 98 women aged 18 to 49 years, vitamin D3 administered at a therapeutic dose of 1250 μg/week for at least 2 months reduced the hs-CRP and homocysteine levels [21]. This might prove helpful in minimizing the risk of cardiovascular disease in overweight women. Another study that included subjects of a variety of races revealed an inverse relationship between homocysteine and 25(OH)D levels among participants with serum levels of 25(OH)D ≤ 21 ng/mL, independent of risk factors for cardiovascular disease or serum levels of vitamin B12 and folate [20]. The Longitudinal Aging Study Amsterdam, a cohort study of advanced-age individuals, revealed a U-shaped relationship between homocysteine and 25(OH)D levels. The lowest homocysteine levels occurred when the 25(OH)D level was between 20 and 24 ng/mL, with the transition point estimated at 20.7 ng/mL [22]. Compared to previous studies, we aimed to identify the relationship between serum homocysteine and vitamin D levels and risk factors for cardiovascular disease only in the healthy Korean population. Compared to women, men had a higher BMI, SBP, DBP, fasting glucose, triglycerides, uric acid, hs-CRP, and homocysteine, and lower vitamin D and HDL-cholesterol. When considering all 1375 participants, we found an inverse linear relationship between homocysteine and vitamin D levels after adjusting for covariates. The mechanism that could explain the association between homocysteine and 25(OH)D concentration is still unclear. One of the important pathways of homocysteine metabolism is transsulfuration, which requires cofactor vitamin B6 and the enzyme cystathionine β-synthase (CBS). This CBS enzyme deficiency can cause hyperhomocysteinemia. Moreover, because CBS is a target gene of the vitamin D receptor, it is thought that homocysteine metabolism may be affected by vitamin D levels [20].

This study found a positive association between homocysteine levels and age and current smoking status in asymptomatic Korean adults. A similar conclusion was reached by Chen et al. [23] in a study of 576 Chinese participants with cardiovascular disease in stable condition. The authors found that homocysteine levels were positively correlated with age and were significantly higher in current smokers than in nonsmokers [23]. A case–control study of 750 patients with vascular disease and 800 control subjects, aged < 60 years in 10 European countries indicated that smokers had an increased risk of vascular disease, and this risk increased significantly with elevated homocysteine. Also, smokers with plasma levels of homocysteine > 12 μmol/L had a 12-fold increased risk of cardiovascular disease compared to nonsmokers with normal homocysteine levels [24]. The mechanism of smoking-induced changes may explain the blockade of the methionine pathway of homocysteine methylation, a methyl donor used to repair cellular damage caused by oxidation produced by cigarette smoking. Therefore, the increase in homocysteine levels due to smoking is considered partly related to the decrease in folate [23,25,26].

We also revealed that serum levels of creatinine and uric acid were positively associated with homocysteine levels. Similarly, a cross-sectional study showed that homocysteine levels were independently and positively associated with serum levels of urea nitrogen and creatinine in advanced-age men with hypertension [9]. Several studies have reported that a lower estimated glomerular filtration rate is associated with elevated homocysteine levels [21,27,28]. A study of 3387 participants aged ≥ 40 years using data from the National Health and Nutrition Examination Survey indicated a strong association between renal insufficiency and increased homocysteine levels, independent of vitamin B status [27]. Imbalances in homocysteine metabolism lead to cellular and molecular damage to many organs via oxidative stress induction, homocysteinylation, and excitotoxicity [9,29,30]. Hyperhomocysteinemia can have detrimental effects on renal tubules, renal corpuscles, and renal arteries, and may lead to secondary renal damage due to other complications, such as hypertension [9,29]. A study of adults in the United States showed that serum levels of uric acid were positively correlated with homocysteine levels and that the association was more significant among those with a lower estimated glomerular filtration rate, lower vitamin B12 level, lower folic acid level, and less alcohol consumption [31]. Kidney function plays an important role in homocysteine metabolism because reduced renal function equates to the poor clearance of homocysteine [32].

This study was the first to investigate the association between serum levels of homocysteine and 25(OH)D and risk factors for cardiovascular disease in the healthy South Korean population. However, several limitations should be noted. First, it was difficult to identify the causality underlying the relationship between the levels of homocysteine and 25(OH)D and risk factors for cardiovascular disease because of the cross-sectional study design. This inherent limitation of the study design restricts the ability to deduce directional relationships. Second, we did not include information related to participants’ geographic location, latitude, or seasonal variations, which are known to influence vitamin D synthesis from sunlight exposure and could potentially impact the observed associations. Third, we did not assess the vitamin B12 or folate levels in our participants, nor did we evaluate the potential impacts of their supplementations. Therefore, we could not investigate the potential interaction of these nutrients with 25(OH)D levels, homocysteine levels, or cardiovascular disease. Fourth, we did not include dietetic information that might potentially impact homocysteine levels and influence the observed associations, nor did we evaluate methylenetetrahydrofolate reductase (MTHFR) polymorphism which determines homocysteine levels among confounders. 

Finally, the study was performed at a single center and might have been subject to selection bias, limiting the generalizability of the findings to wider populations. These factors collectively highlight the need for cautious interpretation of the results and suggest directions for future research to address these gaps.

## 5. Conclusions

The 25(OH)D level was negatively associated with homocysteine levels, whereas age; a history of hypertension; a history of diabetes; current smoking habits; and levels of LDL-cholesterol, creatinine, uric acid, and hs-CRP are positively associated with homocysteine levels in an asymptomatic Korean population. Further long-term prospective studies supporting a causal role of vitamin D status in determining serum levels of homocysteine in asymptomatic Korean adults are needed. In addition, a study evaluating whether lowering the homocysteine level through vitamin D administration in the asymptomatic Korean population can prevent cardiovascular disease might have clinical significance.

## Figures and Tables

**Figure 1 nutrients-16-01155-f001:**
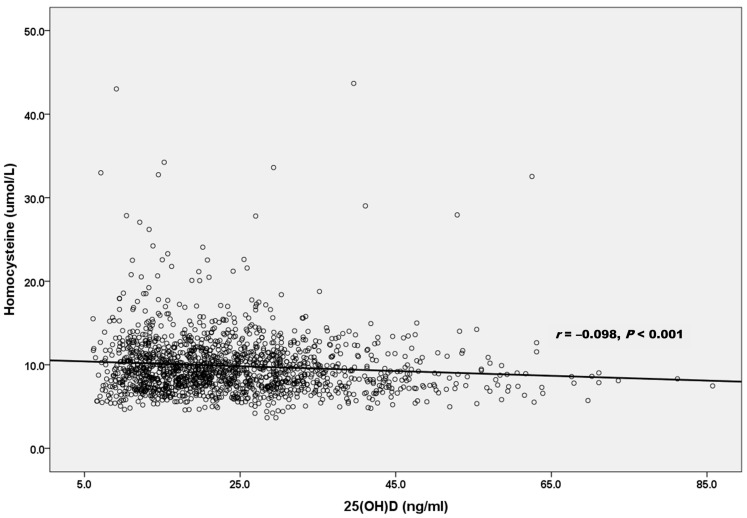
Relationship between 25(OH)D and homocysteine levels.

**Table 1 nutrients-16-01155-t001:** Characteristics of the participants.

Variables		Men (*n* = 895)	Women (*n* = 480)	*p*-Value
Age (y)		53.23 ± 9.59	51.48 ± 10.48	0.002
Body mass index (kg/m^2^)		25.32 ± 3.32	23.20 ± 3.62	<0.001
Waist circumference (cm)		88.49 ± 8.66	79.52 ± 9.65	<0.001
SBP (mmHg)		126.84 ± 13.21	120.71 ± 13.81	<0.001
DBP (mmHg)		79.29 ± 10.24	72.99 ± 9.88	<0.001
Hypertension				<0.001
	Yes	260 (29.1)	76 (15.8)	
	No	635 (70.9)	404 (84.2)	
Diabetes Mellitus				<0.001
	Yes	119 (13.3)	25 (5.2)	
	No	776 (86.7)	455 (94.8)	
Dyslipidemia				0.410
	Yes	127 (14.2)	60 (12.5)	
	No	768 (85.8)	420 (87.5)	
Smoking				<0.001
	Non-smoker	253 (28.3)	459 (95.8)	
	Ex-smoker	263 (29.4)	14 (2.9)	
	Current-smoker	378 (42.3)	6 (1.3)	
Alcohol				<0.001
	No (≤1/week)	532 (59.5)	403 (84.1)	
	Yes (>1/week)	362 (40.5)	76 (15.9)	
Exercise				0.037
	No	167 (18.7)	121 (25.2)	
	1–3 times/week, irregular	421 (47.0)	191 (39.8)	
	1–3 times/week, regular	130 (14.5)	75 (15.6)	
	≥4 times/week	173 (19.3)	91 (19.0)	
Fasting glucose (mg/dL)		104.56 ± 26.85	95.54 ± 19.40	<0.001
HbA1c (%)		5.68 ± 0.87	5.47 ± 0.66	<0.001
Triglycerides (mg/dL)		130.94 ± 77.06	89.63 ± 48.84	<0.001
HDL-cholesterol (mg/dL)		51.32 ± 12.21	63.76 ± 14.09	<0.001
LDL-cholesterol (mg/dL)		118.09 ± 34.52	115.97 ± 29.74	0.233
Creatinine (mg/dL)		0.89 ± 0.06	0.63 ± 0.11	<0.001
Uric acid (mg/dL)		6.15 ± 1.38	4.66 ± 1.01	<0.001
Hs-CRP (mg/dL)		0.14 ± 0.37	0.09 ± 0.13	0.001
TSH (μIU/mL)		2.01 ± 1.36	2.45 ± 1.93	<0.001
25(OH)D (ng/mL)		23.20 ± 10.66	25.02 ± 13.23	0.009
Homocysteine (μmol/L)		10.99 ± 3.89	7.95 ± 2.26	<0.001

Values are means ± standard deviation or numbers (percentages). *p*-values are obtained with the Student’s *t*-test or chi-square test. SBP, systolic blood pressure; DBP, diastolic blood pressure; HbA1c, glycated hemoglobin; Hs-CRP, high-sensitive C-reactive protein; LDL, low-density lipoprotein; HDL, high-density lipoprotein; TSH, thyroid-stimulating hormone; 25(OH)D, 25-hydroxyvitamin D.

**Table 2 nutrients-16-01155-t002:** Correlations between homocysteine level and clinical variables.

Variables	*r*	*p*-Value
Age (y)	0.077	0.004
Body mass index (kg/m^2^)	0.110	<0.001
Waist circumference (cm)	0.163	<0.001
SBP (mmHg)	0.104	<0.001
DBP (mmHg)	0.121	<0.001
Fasting glucose (mg/dL)	0.076	0.005
HbA1c (%)	0.072	0.008
Triglycerides (mg/dL)	0.138	<0.001
HDL-cholesterol (mg/dL)	−0.142	<0.001
LDL-cholesterol (mg/dL)	0.072	0.008
Creatinine (mg/dL)	0.473	<0.001
Uric acid (mg/dL)	0.310	<0.001
Hs-CRP (mg/dL)	0.088	0.001
TSH (μIU/mL)	−0.037	0.175
25(OH)D (ng/mL)	−0.098	<0.001

*p*-values were obtained by Pearson’s correlation tests. SBP, systolic blood pressure; DBP, diastolic blood pressure; HbA1c, hemoglobin A1c; Hs-CRP, high-sensitive C-reactive protein; LDL, low-density lipoprotein; HDL, high-density lipoprotein; TSH, thyroid-stimulating hormone; 25(OH)D, 25-hydroxyvitamin D.

**Table 3 nutrients-16-01155-t003:** Associations between serum levels of homocysteine and 25(OH)D.

	*β*	*p*-Value
Model 1	−0.027	0.001
Model 2	−0.025	0.003
Model 3	−0.033	<0.001

Model 1: with adjustments for age and sex. Model 2: Model 1 with adjustments for body mass index, waist circumference, systolic blood pressure, diastolic blood pressure, smoking, alcohol drinking, and exercise. Model 3: Model 2 with adjustments for the presence of hypertension, diabetes mellitus, dyslipidemia, fasting glucose, hemoglobin A1c, triglycerides, high-density lipoprotein-cholesterol, low-density lipoprotein-cholesterol, creatinine, uric acid, high-sensitive C-reactive protein, and thyroid-stimulating hormone.

**Table 4 nutrients-16-01155-t004:** Multiple linear regression analysis of homocysteine levels and clinical variables in all participants (*n* = 1375).

Variables		*β*	*p*-Value
Age (y)		0.025	0.016
Sex (Women)		−0.767	0.014
Body mass index (kg/m^2^)		−0.038	0.438
Waist circumference(cm)		−0.019	0.298
SBP (mmHg)		−0.001	0.930
DBP (mmHg)		0.003	0.794
Hypertension		0.876	<0.001
Diabetes Mellitus		1.480	<0.001
Dyslipidemia		0.053	0.846
Smoking			
	Ex-smoker	0.123	0.630
	Current smoker	0.892	0.001
Alcohol		−0.157	0.435
Exercise			
	1–3 times/week, irregular	0.044	0.879
	1–3 times/week, regular	−0.349	0.127
	≥4 times/week	−0.541	0.051
Fasting glucose (mg/dL)		−0.002	0.694
HbA1c (%)		−0.223	0.206
Triglycerides (mg/dL)		0.001	0.646
HDL-cholesterol (mg/dL)		0.014	0.059
LDL-cholesterol (mg/dL)		0.009	0.002
Creatinine (mg/dL)		6.788	<0.001
Uric acid (mg/dL)		0.245	0.002
Hs-CRP (mg/dL)		0.687	0.016
TSH (μIU/mL)		0.039	0.483
25(OH)D (ng/mL)		−0.033	<0.001

SBP, systolic blood pressure; DBP, diastolic blood pressure; HbA1c, glycated hemoglobin; Hs-CRP, high-sensitive C-reactive protein; LDL, low-density lipoprotein; HDL, high-density lipoprotein; TSH, thyroid-stimulating hormone; and 25(OH)D, 25-hydroxyvitamin D.

## Data Availability

The data are available upon reasonable request due to ethical restrictions on data sharing.

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
