# Peer review of "Association between Homocysteine and Vitamin D Levels in Asymptomatic Korean Adults"

_nutrients, 2024, doi:10.3390/nu16081155_

Round 1

Reviewer 1 Report

Comments and Suggestions for Authors

This fascinating cross-sectional study shows a negative association between the serum concentration of homocysteine and that of 25-hydroxyvitamin D [25(OH)D]. Because of the association between elevated homocysteine serum concentrations and various cardiovascular problems, this association could well be related to a benefit of good vitamin D status and the lower risk of clinical cardiovascular abnormalities.

There is one problem that is pervasive throughout the text and that is the use of the term “serum vitamin D levels”. This would be fully understandable as a shorthand term for vitamin D status. However, what was actually measured in serum was not vitamin D, but rather the vitamin D metabolite, 25(OH)D. Both vitamin D and 25(OH)D are present in most serum samples. Therefore, it is confusing to use the term “vitamin D levels” as that could literally mean the concentration of vitamin D in serum rather than the concentration of 25(OH)D.

The misuse of the term “vitamin D level” needs correcting in the following lines: 57, 59, 69, 154, 155, 157, 159, 163, 197, 201, 204, 206, 208, 259, 262, 269, 275. The term “Vitamin D (ng/mL)” in Tables 1, 2 and 4 also needs correcting. The same problem occurs in the X axis of Figure 1 and also in the legend of Figure 1.

Author Response

We thank the reviewers for their valuable opinions and sincere and detailed comments. We submit a revised version of the manuscript that includes modifications and amendments based on the reviewers’ opinions.

The following are responses to each of the reviewers’ comments:

# Reviewer 1.

Comments and Suggestions for Authors:

This fascinating cross-sectional study shows a negative association between the serum concentration of homocysteine and that of 25-hydroxyvitamin D [25(OH)D]. Because of the association between elevated homocysteine serum concentrations and various cardiovascular problems, this association could well be related to a benefit of good vitamin D status and the lower risk of clinical cardiovascular abnormalities.

There is one problem that is pervasive throughout the text and that is the use of the term “serum vitamin D levels”. This would be fully understandable as a shorthand term for vitamin D status. However, what was actually measured in serum was not vitamin D, but rather the vitamin D metabolite, 25(OH)D. Both vitamin D and 25(OH)D are present in most serum samples. Therefore, it is confusing to use the term “vitamin D levels” as that could literally mean the concentration of vitamin D in serum rather than the concentration of 25(OH)D.

The misuse of the term “vitamin D level” needs correcting in the following lines: 57, 59, 69, 154, 155, 157, 159, 163, 197, 201, 204, 206, 208, 259, 262, 269, 275. The term “Vitamin D (ng/mL)” in Tables 1, 2 and 4 also needs correcting. The same problem occurs in the X axis of Figure 1 and also in the legend of Figure 1.

Answer) Thank you for your detailed and sincere comments. As you suggested, we changed the wording of “vitamin D level” to “25(OH)D” in the following lines: 57, 59, 69, 154, 155, 157, 159, 163, 197, 201, 204, 206, 208, 259, 262, 269, 275, tables 1, 2, 4, figure 1, and the legend of figure 1. The “25(OH)D” is considered to be an appropriate expression. Thank you for your valuable opinions.

Reviewer 2 Report

Comments and Suggestions for Authors

The authors submitted a retrospective study on 1,375 healthy Korean adults who were outpatients in a Health promotion centre between 2018 and 2022. The authors investigated the association between homocysteine levels and anthropometric and health variables. In particular, they highlighted the cross-sectional relationship between vitamin D and homocysteine levels in an adjusted multivariate analysis and found a significant relationship.

The manuscript is well-organized with adequate references. The text is fluent and the methods are described in detail.

Unfortunately, the study's design and the association's low value make it difficult to extrapolate a clinical meaning. However, this work will suggest a field of research that deserves more attention.

Some aspects could need further attention:

In the description of participants, BMI (according to est-asian cutoffs), waist circumference and SBP seem borderline. Do the authors think these characteristics may have influenced results?

Vitamin D and HCY concentrations seem within adequate levels. Do the authors consider this aspect a limit to detect a stronger association?

The limitations of the study are fair but the authors must include the missing dietetics information and MTHFR polymorphism among confounders. The former is very important considering how much different diets can influence HCY levels and their interpretation (https://doi.org/10.1093/clinchem/47.6.1094). The latter is very important because of the strong influence of MTHFR polymorphisms on circulating HCY levels (https://doi.org/10.1111/jch.13737).

Some unexpected links between essential nutrients and hcy can rise in literature, such as in the case of PUFAs and HCY (https://doi.org/10.3390/biom10020219). I suggest that the authors propose a possible mechanism to explain the association between vitamin D and HCY levels.

Author Response

We thank the reviewers for their valuable opinions and sincere and detailed comments. We submit a revised version of the manuscript that includes modifications and amendments based on the reviewers’ opinions.

The following are responses to each of the reviewers’ comments:

# Reviewer 2.

Comments and Suggestions for Authors:

The authors submitted a retrospective study on 1,375 healthy Korean adults who were outpatients in a Health promotion centre between 2018 and 2022. The authors investigated the association between homocysteine levels and anthropometric and health variables. In particular, they highlighted the cross-sectional relationship between vitamin D and homocysteine levels in an adjusted multivariate analysis and found a significant relationship.

The manuscript is well-organized with adequate references. The text is fluent and the methods are described in detail.

Unfortunately, the study's design and the association's low value make it difficult to extrapolate a clinical meaning. However, this work will suggest a field of research that deserves more attention.

Some aspects could need further attention:

In the description of participants, BMI (according to est-asian cutoffs), waist circumference and SBP seem borderline. Do the authors think these characteristics may have influenced results?

Answer) Thank you for your considerate and valuable opinions. We described the significant differences between genders in several areas including BMI, waist circumference, and SBP, which seem borderline. Because we used multiple linear regression analysis to assess the association between serum homocysteine and 25(OH)D levels and adjusted for various covariates, including BMI, waist circumference, and SBP, these characteristics may not have influenced the results. We sincerely appreciate your valuable comments.

Vitamin D and HCY concentrations seem within adequate levels. Do the authors consider this aspect a limit to detect a stronger association?

Answer) Thank you for your considerate and sincere comments. As you mentioned, 25(OH)D and homocysteine concentrations, which are expressed as means ± standard deviation in Table 1, seem within adequate levels. However, we do not think that this aspect will limit the detection of stronger associations between the two. We evaluated the association between homocysteine and 25(OH)D levels in the healthy asymptomatic Korean population. Considering all 1,375 participants, multivariate linear regression analysis revealed an inverse relationship between homocysteine and 25(OH)D levels after adjusting for covariates (β = 0.033, P < 0.001). Additional studies investigating the relationship between 25(OH)D and homocysteine in patients with cardiovascular disease should be considered and compared with our findings.

The limitations of the study are fair but the authors must include the missing dietetics information and MTHFR polymorphism among confounders. The former is very important considering how much different diets can influence HCY levels and their interpretation (https://doi.org/10.1093/clinchem/47.6.1094). The latter is very important because of the strong influence of MTHFR polymorphisms on circulating HCY levels (https://doi.org/10.1111/jch.13737).

Answer) Thank you so much for your sincere and detailed suggestions, references, and valuable opinions. As you mentioned, we revised the manuscript and the details have been added to the limitation part of this study in the Discussion section as shown below.

"Fourth, we did not include dietetic information that might potentially impact homocysteine levels and influence the observed associations, nor did we evaluate methylenetetrahydrofolate reductase (MTHFR) polymorphism which determines homocysteine levels among confounders."

Some unexpected links between essential nutrients and hcy can rise in literature, such as in the case of PUFAs and HCY (https://doi.org/10.3390/biom10020219). I suggest that the authors propose a possible mechanism to explain the association between vitamin D and HCY levels.

Answer) Thank you so much for your valuable and considerate suggestions. As you mentioned, the details related to a possible mechanism to explain the association between homocysteine and 25(OH)D levels have been noted in the Discussion section and highlighted.

"The mechanism that could explain the association between homocysteine and 25(OH)D concentration is still unclear. One of the important pathways of homocysteine metabolism is transsulfuration, which requires cofactor vitamin B6 and the enzyme cystathionine β-synthase (CBS). This CBS enzyme deficiency can cause hyperhomocysteinemia. Moreover, because CBS is a target gene of the vitamin D receptor, it is thought that homocysteine metabolism may be affected by vitamin D levels [20]."

Round 2

Reviewer 2 Report

Comments and Suggestions for Authors

The authors adequately answered my questions. No further requests